# The Rnf complex is a Na$^+$ coupled respiratory enzyme in a fermenting bacterium, *Thermotoga maritima*

Martin Kuhns[1,3], Dragan Trifunović[1,3], Harald Huber[2] & Volker Müller [1✉]

*rnf* genes are widespread in bacteria and biochemical and genetic data are in line with the hypothesis that they encode a membrane-bound enzyme that oxidizes reduced ferredoxin and reduces NAD and vice versa, coupled to ion transport across the cytoplasmic membrane. The Rnf complex is of critical importance in many bacteria for energy conservation but also for reverse electron transport to drive ferredoxin reduction. However, the enzyme has never been purified and thus, ion transport could not be demonstrated yet. Here, we have purified the Rnf complex from the anaerobic, fermenting thermophilic bacterium *Thermotoga maritima* and show that is a primary Na$^+$ pump. These studies provide the proof that the Rnf complex is indeed an ion (Na$^+$) translocating, respiratory enzyme. Together with a Na$^+$-F$_1$F$_O$ ATP synthase it builds a simple, two-limb respiratory chain in *T. maritima*. The physiological role of electron transport phosphorylation in a fermenting bacterium is discussed.

[1] Molecular Microbiology & Bioenergetics, Institute of Molecular Biosciences, Johann Wolfgang Goethe University Frankfurt Main, Max-von-Laue-Str. 9, 60438 Frankfurt, Germany. [2] Department of Microbiology, University of Regensburg, Universitätsstraße 31, 93053 Regensburg, Germany. [3] These authors contributed equally: Martin Kuhns, Dragan Trifunović. ✉email: vmueller@bio.uni-frankfurt.de

Biological energy conservation in a primordial, anoxic world most likely involved separation of charges and molecules such as protons and sodium ions across membranes[1] and the established electrochemical gradient as driving force for the synthesis of ATP, the universal energy currency in every living cell, by simple ATP synthases[2]. How these gradients have been established is still an enigma, but anaerobic microbes may shed light on this question. They often employ ferredoxin (Fd) in addition to NAD as electron carrier. The low potential electron carrier ferredoxin ($E_O' = -450$ to $-500$ mV[3]) can be reduced by only a few enzymes such as glycerol aldehyde 3-phosphate dehydrogenase[4] or pyruvate:ferredoxin oxidoreductase[5], both employed during chemoorganoheterotrophic growth (glycolysis). During the chemolithoautotrophic growth, ferredoxin must be reduced with $H_2$ ($E_O' = -414$ mV) as electron donor, but this reaction is highly endergonic. The energy barrier is overcome by flavin-based electron bifurcation, a recently established mechanism widespread in anaerobes to drive endergonic ferredoxin reduction by coupling the oxidation of the same substrate to the reduction of a second, more electropositive acceptor[6,7]. In the electron-bifurcating hydrogenases the second electron acceptor is NAD[8,9].

$Fd_{red}$ is an ideal electron donor in an anoxic world since, due to its low redox potential, electrons can be delivered to acceptors such as, for example, protons ($E_O' = -414$ mV) or NAD ($E_O' = -320$ mV). Reduction of protons and NAD with $Fd_{red}$ as donor is exergonic and nature has evolved different enzymes that store the redox energy in a transmembrane electrochemical ion potential across the cytoplasmic membrane[10–13]. $Fd_{red}$:$H^+$ oxidoreductase activity coupled to proton translocation is found in some anaerobic archaea[11,14,15] and has been studied to a great extent in *Pyrococcus furiosus*[12]. Experiments with inverted membrane vesicles clearly demonstrated this respiratory activity, and the enzyme catalyzing this reaction, the membrane-bound hydrogenase (Mbh) has been purified and characterized[16]. It is a predecessor of complex I and the recently solved structure of the enzyme revealed its structural similarity to complex I[17]. Although the enzyme has been purified, it has not been reconstituted yet in liposomes. Therefore, the final proof that the enzyme is a proton pump is still missing.

The second $Fd_{red}$-dependent electron transport chain found in anaerobes uses $NAD^+$ as electron acceptor. $Fd^{2-}$:$NAD^+$-oxidoreductase activity was demonstrated in the acetogen *Acetobacterium woodii*[18] and inverted membrane vesicles of this bacterium coupled $Fd^{2-}$:$NAD^+$-oxidoreductase to vectorial $Na^+$ transport across the cytoplasmic membrane[10]. The activity was enriched from membranes of *A. woodii* and the preparation contained, amongst others, proteins with similarity to proteins encoded by the *rnf* genes[19]. These genes have been firstly described in and shown to be essential for $N_2$ fixation in *Rhodobacter capsulatus*[20]. The *rnf* genes are assumed to encode a membrane-bound enzyme complex with six subunits; two subunits are predicted to be membrane integral, the other four are cytosolic. RnfB is predicted to be the entry point for electrons coming from $Fd_{red}$[21], RnfC is the predicted NAD binding site[21]. Electron flow from RnfB to RnfC is most likely via RnfA and RnfG. Subunits D and E are membrane-integral subunits suggested to mediate ion ($Na^+$) transport, driven by electron transport that involves iron sulfur cluster, monovalent iron and covalently bound flavins. Again, our view of the structure and function of Rnf complexes stems from predictions of DNA sequences and analogies to the well-studied respiratory enzyme Nqr[22,23]. The final proof that the purified Rnf complex is a respiratory enzyme is still missing, largely due to the fact that the complex always escaped purification, even from different microbial sources[24].

We describe here a procedure to purify the Rnf complex from a thermophilic bacterium, *Thermotoga maritima*, in complex with the ATP synthase. The purified Rnf-ATP synthase supercomplex was reconstituted into liposomes and its catalytic properties were analyzed. The Rnf complex was further separated from the ATP synthase and shown to catalyze $Na^+$ transport.

## Results

**Purification of Rnf and ATP synthase.** Since attempts to purify the Rnf complex from *A. woodii*, *Clostridium ljungdahlii* or *Clostridium tetanomorphum* failed due to an inherent instability of the complex,[24] we rationalized that the complex from a thermophilic bacterium may be more stable. Inspection of the genome sequence of *T. maritima* ($T_{opt} = 80$ °C) revealed the presence of *rnf* genes in the order *rnfCDGEAB*, as in *A. woodii*. The predicted proteins are 46.9, 46.9, 29.3, 59.8, 56.1 and 49.2% similar to the corresponding proteins from *A. woodii*. Interestingly, RnfB is only 10.56 kDa, compared to 37.73 kDa in *A. woodii*. This accomodates only one predicted 4Fe-4S center, in contrast to six predicted 4Fe-4S in *A. woodii*. This raised the question whether the Rnf complex from *T. maritima* would accept ferredoxin from *Clostridium pasteurianum*, which is generally used as electron donor to measure $Fd_{red}$:NAD-oxidoreductase activity. $Fd_{Cp}$ was purified and reduced with carbon monoxide, catalyzed by the purified carbon monooxide dehydrogenase from the mesophile *A. woodii*. Much to our relief, membranes of *T. maritima* catalyzed $Fd_{red}$-dependent $NAD^+$ reduction at 60 °C, a temperature that meets the demands of the auxilliary proteins (CO dehydrogenase ($T_{opt} = 30$ °C), ferredoxin ($T_{opt} = 37$ °C)) from the mesophile as well as the target enzyme, Rnf, from a thermophile. The $Fd^{2-}$:$NAD^+$-oxidoreductase activity at membranes of *T. maritima* was 97 mU/mg, measured at 60 °C (Supplementary Fig. 1).

The enzyme catalyzing $Fd^{2-}$:$NAD^+$-oxidoreductase activity was purified from the cells of *T. maritima* under strictly anoxic conditions. Solubilization was achieved with *n*-Dodecyl *β*-D-maltoside (1 mg/mg protein), followed by anion exchange chromatography on Q-sepharose and gel filtration on Sephacryl S300. The Rnf activity eluted as a single peak from a gel filtration with an activity of 2.1 U/mg (Supplementary Fig. 2). This procedure enriched the activity 21-fold with a yield of 13% (Supplementary Table 1). When the preparation was analyzed by SDS-PAGE, more than the expected six proteins were visible (Supplementary Fig. 3a). RnfD (34.6 kDa) and RnfG (24.3 kDa) have covalently bound flavins and indeed, two proteins with apparent masses of 23 and 32 kDa were visible after exposure of the gel to UV light (Supplementary Fig. 3b). Antibodies against Rnf subunits from *A. woodii* did not cross react with any protein in the preparation which is plausible since the identities are only between 29.3 and 59.8%. In a native PAGE, several bands were visible. To determine the Rnf complex, we did an in-gel assay to determine the NADH:MTT-oxidoreductase activity. Two complexes of apparent masses of ≈150 and ≈290 kDa had NADH:MTT-oxidoreductase activity and the activity of the ≈290-kDa complex was higher than in the 150-kDa complex (Fig. 1a). The predicted mass of an Rnf monomer (one subunit per complex) is 160 kDa; thus the two complexes could represent a monomer and a dimer of the Rnf complex, which is within the error of the mass detection (9%) by native PAGE and the uncertainty of the subunit stochiometries. The ≈290-kDa complex was excised from the gel and applied to a SDS-PAGE. As evident from Fig. 1b, proteins with masses matching the predicting masses of the Rnf complex were visible. LC/MS/MS and MALDI-TOF analyses revealed three of six Rnf subunits (RnfCDG) in the preparation (Supplementary Table 2).

The native gel contained a 550-kDa complex of high abundance. This mass corresponds to the mass of the $F_1F_O$-ATP synthase encoded in the genome of *T. maritima* (545–569 kDa, depending on the number of *c* subunits in the complex).

Indeed, the complex had in-gel ATPase activity (Fig. 1a). The complex was excised and separated in a SDS-PAGE. Proteins with masses expected for subunits $\alpha$, $\beta$, $\gamma$, $\delta$, $\varepsilon$, $a$, $c$ and $b$ were detected by silver staining (Fig. 1b). Indeed, the 60- and 8-kDa proteins reacted with antibodies against subunits $\beta$ and $c$ of the $F_1F_O$-ATP synthase of *A. woodii*. Moreover all eight subunits were detected by LC/MS/MS and MALDI-TOF (Supplementary Table 2). The ATPase activity in the preparation was around 5 U/mg.

**The Rnf complex requires Na$^+$ and is inhibited by DCCD.** $Fd^{2-}$:NAD$^+$-oxidoreductase activity was similar in 50 mM Tris-HCl (pH 7.7), 50 mM 3-(N-morpholino)propanesulfonic acid (MOPS) (pH 7.7) and 50 mM 1,4-Piperazinediethanesulfonic acid (PIPES) buffer (pH 7.7) and somewhat lower in 50 mM TES and 50 mM K-PO$_4$ buffer. For the following, 50 mM Tris-HCl was used. Activity had a rather broad pH optimum from 7 to 8.5, but at pH 6 and 10.5, activity was still 50%. The temperature optimum could not be determined using the physiological activity since CODH used to reduce ferredoxin is from a mesophile and is inactive at temperatures

>70 °C. Therefore, NADH:MV-oxidoreductase activity was measured. As seen in Supplementary Fig. 4, activity increased with increasing temperatures with a maximum of 51.4 ± 2.4 U/mg at the highest temperature tested, 90 °C. At 60 °C, still 47.1% of the activity was present. At 60 °C, the activity of the CODH was still 35%. Therefore, the $Fd^{2-}$:NAD$^+$-oxidoreductase activity was measured at the compromise temperature of 60 °C.

Of special importance was the effect of Na$^+$ on $Fd^{2-}$:NAD$^+$-oxidoreductase since Na$^+$- and H$^+$-dependent Rnf complexes are known. In the absence of Na$^+$ there was no $Fd^{2-}$:NAD$^+$-oxidoreductase activity. KCl did not stimulate activity, addition of LiCl stimulated activity slightly, but addition of NaCl greatly stimulated the activity. The dependence of activity on the NaCl concentration followed a Michaelis−Menten kinetic with a $K_m$ of around 0.5 mM and a $v_{max}$ of 1 U/mg (Fig. 2a). These data clearly show that the Rnf complex of *T. maritima* is Na$^+$-dependent. Some membrane proteins involved in ion transport are inhibited by *N,N′*-dicyclohexylcarbodiimide (DCCD), which binds covalently to carboxylates buried in the membrane phase[25,26]. DCCD also

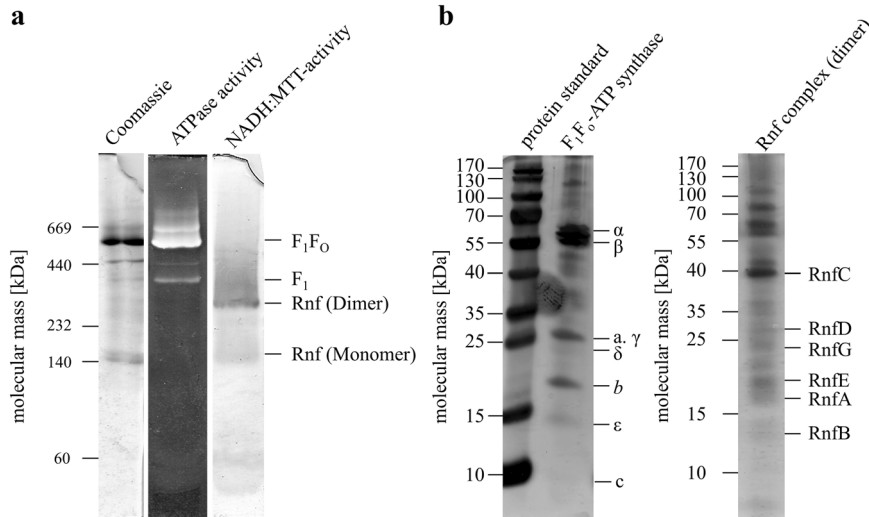

**Fig. 1 Separation and identification of subunits of the Rnf complex and F$_1$F$_O$ ATP synthase from *T. maritima*. a** Twenty micrograms of purified protein was separated on a native-PAGE[49]. The proteins were visualized by staining with Coomassie Brilliant Blue. ATPase and Rnf activity were determined as described in "Methods". **b** The bands representing Rnf (dimer) and F$_1$F$_O$ were excised from the native gel and separated on a 12.5% SDS-gel according to ref. [48] and stained with silver. The subunits were identified by western blot or by LC/MS/MS and MALDI-TOF.

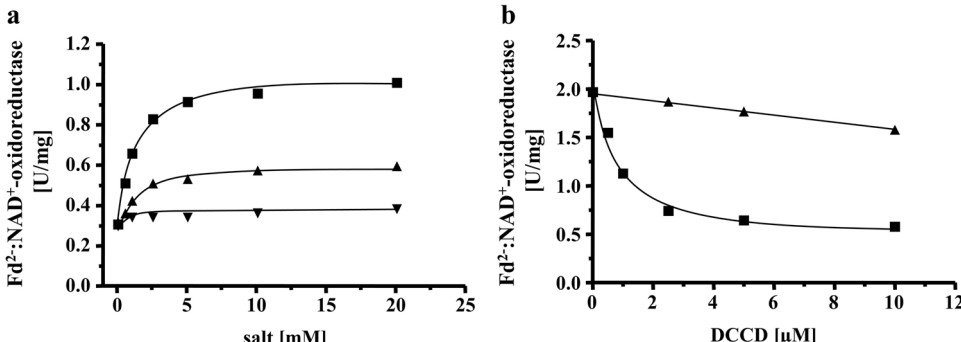

**Fig. 2 Na$^+$ stimulates ferredoxin:NAD$^+$ oxidoreductase activity and protects from DCCD inhibition. a** Purified protein (7.5 μg) was added to 1 ml buffer (20 mM Tris-HCl, 2 mM DTE, 2.2 μM resazurin, pH 7.7) and ferredoxin:NAD$^+$ oxidoreductase was measured as described in "Methods". Increasing amounts (0−20 mM) of NaCl (solid square), KCl (inverted solid triangle) or LiCl (solid triangle) were applied to the assays. The contaminating Na$^+$ concentration was 0.095 mM. **b** Purified protein (10 μg) was added to 1 ml buffer (20 mM Tris-HCl, 2 mM DTE, 2.2 μM resazurin, pH 7.7) with increasing amounts of DCCD (0–10 μM) in the presence (solid triangle) or absence (solid square) of 20 mM NaCl. The samples were incubated for 20 min at room temperature followed by an incubation at 60 °C for 5 min before the measurement started. If NaCl was omitted during preincubation, 20 mM NaCl was added before the start of the reaction. The contaminating Na$^+$ concentration was 0.12 mM. These data points represent two different biological replicates.

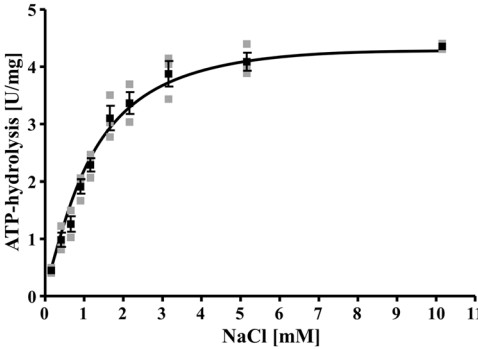

**Fig. 3 Na⁺ dependence of ATP hydrolysis.** Four micrograms of purified ATPase was added to 1200 μl ATPase buffer (100 mM Tris-HCl, 100 mM malic acid, pH 7.4) containing 0−10 mM NaCl. The sample was incubated for 5 min at 80 °C. The reaction was started by adding 3 mM K₂-ATP. The contaminating Na⁺ concentration was 0.154 mM. All data points are mean ± SEM; N = 3 independent experiments.

inhibited Fd²⁻:NAD⁺-oxidoreductase; maximum inhibition was observed at 5 μM DCCD. Most interestingly, DCCD inhibition was relieved by the addition of Na⁺ (Fig. 2b). When the enzyme was preincubated with 5 μM DCCD and 20 mM Na⁺, DCCD inhibition was only 20%. Li⁺ also protected from DCCD inhibition, but to a much smaller extent. At 5 μM DCCD and 20 mM LiCl, the protection was only 21%. KCl did not protect the enzyme from DCCD inhibition (Supplementary Fig. 5). The different starting activities are due to variations in activity of different badges of enzyme. These data indicate that, as in other Na⁺ translocating membrane proteins such as the Na⁺-F₁F₀-ATP synthase, Na⁺ and DCCD compete for a common binding site[27,28].

**The ATP synthase of *T. maritima* is of the Na⁺-F₁F₀-type.** ATP hydrolysis was strictly dependent on the Na⁺ concentration (Fig. 3) and Na⁺ dependence of activity followed a Michalies −Menten kinetic; the $K_m$ was 1.2 ± 0.2 mM Na⁺ and $v_{max}$ was 4.7 U/mg. As seen before with other Na⁺-F₁F₀-ATP synthases[27,28], DCCD inhibited ATP hydrolysis but inhibition was relieved by addition of Na⁺, indicating that Na⁺ and DCCD compete for a common binding site (Fig. 4).

**Reconstitution of ATP synthase and Rnf in liposomes.** To address ion transport by the Rnf complex and the ATP synthase, the enzymes were reconstituted into liposomes. Since the *E. coli* lipids used to prepare liposomes are unstable at higher temperatures, the activity assays were performed at 45 °C. At this temperature, Fd²⁻:NAD⁺-oxidoreductase and ATPase activity were 65% and 55%, respectively, of the control at 60 or 80 °C. At 45 °C the liposomes were quite stable and could hold an artificial pH created by an ammonium diffusion potential and measured by ACMA quenching for 7 min. After incorporating the enzymes by detergent removal, the activities of both were measured: ATPase was 2.6 U/mg and Fd²⁻:NAD⁺-oxidoreductase was 1.2 U/mg. To determine Na⁺ transport, the proteoliposomes were incubated with ²²Na⁺. Upon addition of K₂-ATP, proteoliposomes accumulated Na⁺ in the lumen more than twofold, which clearly demonstrates that the F₁F₀-ATP synthase of *T. maritima* uses Na⁺ as coupling ion (Fig. 5a). When proteoliposomes were incubated with ferredoxin and CODH under a CO atmosphere, ²²Na⁺ was accumulated after addition of the electron acceptor, NAD (Fig. 5b). Accumulation was 2.5-fold, demonstrating that the Rnf complex also uses Na⁺ as coupling ion.

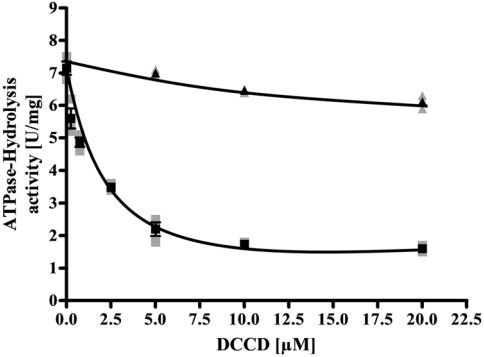

**Fig. 4 Na⁺ protects ATP-hydrolysis from inhibition by DCCD.** Nine micrograms of purified ATPase was added to 1200 μl ATPase buffer (100 mM Tris-HCl, 100 mM malic acid, pH 7.4) containing 0−20 μM DCCD in the absence (solid square) or presence of 20 mM NaCl (solid triangle). The sample was incubated for 30 min at room temperature followed by 5 min at 80 °C. The reaction was started by adding 3 mM K₂-ATP. The contaminating Na⁺ concentration was 0.07 mM. If NaCl was omitted during preincubation, 20 mM NaCl was added before the start of the reaction. All data points are mean ± SEM; N = 3 independent experiments.

**Further purification and reconstitution of the Rnf.** The experiments described so far clearly demonstrate a Na⁺ pumping Rnf and a Na⁺ pumping ATP synthase present in *T. maritima* that co-purify. Since Na⁺ transport with a purified Rnf complex had never been shown, it was important to unequivocally demonstrate Na⁺ transport by the Rnf complex. Therefore, we decided to modify the isolation procedure to remove the ATP synthase. To this end, the solubilisate was precipitated with PEG6000, the precipitate applied to a sucrose density gradient, followed by chromatography on Q-sepharose, hydroxyapatite and sephacryl S300. This procedure removed the Na⁺-F₁F₀-ATP synthase and enriched the Rnf activity by 75%. However, the yield was only 4.1% and the procedure was laborious that often led to inactivation during the purification procedure which usually takes 7−10 days. Anyway, ATPase activity could not be detected in the preparation. The preparation contained all subunits of the Rnf complex, which were confirmed by MALDI-TOF-analyses (Supplementary Table 3) and SDS-PAGE (Supplementary Fig. 6a). RnfD and RnfG contained covalently bound flavins (Supplementary Fig. 6b). In a native PAGE a rather broad band ranging from 215 to 270 kDa was observed. Whether this represent a monomer or a dimer needs to be addressed in the future. Anyway, the complex had NADH:MTT in-gel activity (Supplementary Fig. 6c). Fd²⁻:NAD⁺-oxidoreductase was strictly Na⁺ dependent, inhibited by DCCD and Na⁺ protected from DCCD inhibition (Fig. 6). The Rnf complex was then incorporated into liposomes, as before. When incubated in the presence of ²²Na⁺, cells accumulated ²²Na⁺ when the Fd²⁻:NAD⁺-oxidoreductase activity was started by addition of the electron acceptor NAD (Fig. 7). Accumulation of ²²Na⁺ was inhibited by the Na⁺ ionophore ETH2120, but stimulated by the protonophore TCS, demonstrating that Na⁺ transport is a primary and electrogenic event (Fig. 8).

## Discussion

*rnf* genes are widespread in bacteria and also present in some archaea[18]. Based on the homology of some subunits of the hypothetical Rnf complex to subunits of the NADH:quinone-oxidoreductase (Nqr[29]), it was speculated that the Rnf complex is a respiratory enzyme that oxidizes reduced ferredoxin

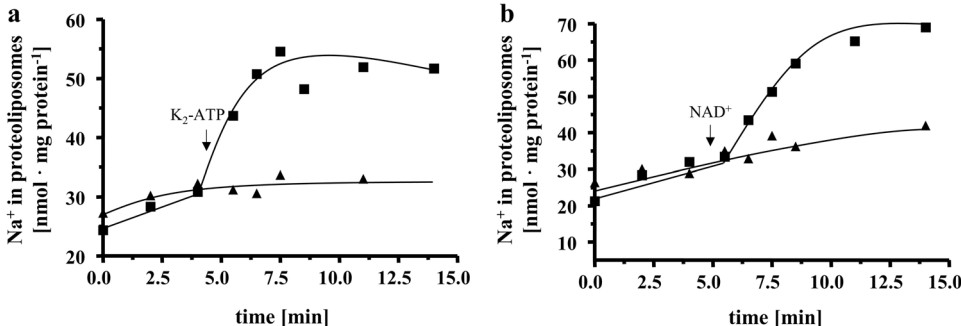

**Fig. 5 $^{22}$Na$^+$ transport by the ATPase and the Rnf complex reconstituted into liposomes. a** Proteoliposomes (protein concentration 1.3 mg/ml) in 100 mM Tris-HCl, 100 mM malic acid, pH 7.4, containing 5 mM MgCl$_2$, 5 mM $^{22}$NaCl, 2 mM DTE, 4 μM resazurin catalyzed $^{22}$Na$^+$ transport upon addition of 15 μl K$_2$-ATP (100 mM) under oxic conditions (solid square). $^{22}$Na$^+$ transport was not observed in the absence of K$_2$-ATP (solid triangle). **b** The same proteolipsomes catalyzed $^{22}$Na$^+$ transport upon addition of 10 μl ferredoxin (3 mM), 10 μg CODH (3 mg/ml) and 30 μl NAD$^+$ (100 mM) and a CO atmosphere (solid square). $^{22}$Na$^+$ transport was not observed in the absence of NAD$^+$ (solid triangle). These data points represent two different biological replicates.

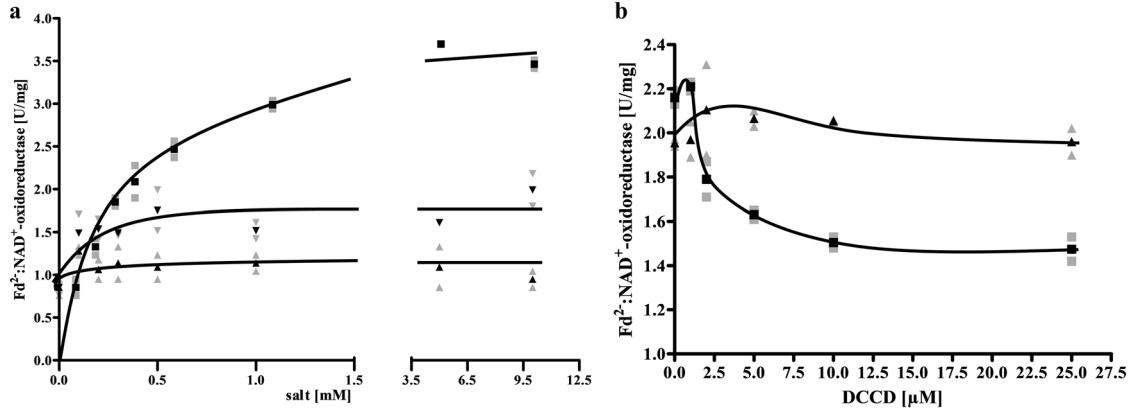

**Fig. 6 Na$^+$ stimulates the ferredoxin:NAD$^+$ oxidoreductase activity catalyzed by the purified Rnf complex and protects from DCCD inhibition.**
**a** Purified protein (10 μg) was added to 1 ml buffer (20 mM Tris-HCl, 2 mM DTE, 2.2 μM resazurin, pH 7.7) and ferredoxin:NAD$^+$ oxidoreductase activity was measured as described in "Methods". Increasing amounts (0−10 mM) of NaCl (solid square), KCl (solid triangle) or LiCl (solid inverted triangle) was applied to the assays. The contaminating Na$^+$ concentration was 0.085 mM. **b** Purified protein (10 μg) was added to 1 ml buffer (20 mM Tris-HCl, 2 mM DTE, 2.2 μM resazurin, pH 7.7) with increasing amounts of DCCD (0–25 μM) in the presence (solid triangle) or absence (solid square) of 20 mM NaCl. The samples were incubated for 20 min at room temperature followed by an incubation at 60 °C for 5 min before the measurement was started. When NaCl was omitted in the preincubation, 20 mM NaCl was added with the start of the reaction. The contaminating Na$^+$ concentration was 0.085 mM. These data points represent two different biological replicates.

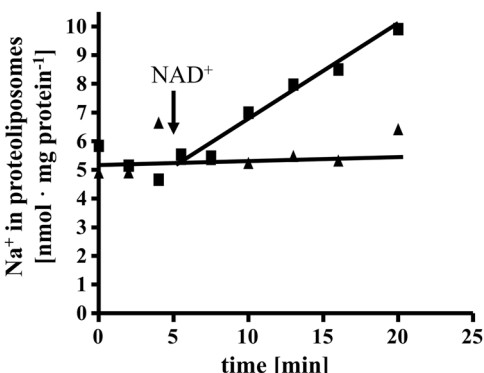

**Fig. 7 $^{22}$Na$^+$ transport by the purified Rnf complex reconstituted into liposomes.** Proteoliposomes (protein concentration 1.06 mg/ml) in 1 ml 100 mM Tris-HCl, 100 mM malic acid, pH 7.4, containing 5 mM MgCl$_2$, 2 mM $^{22}$NaCl, 2 mM DTE, 4 μM resazurin catalyzed $^{22}$Na$^+$ transport upon addition of 10 μl ferredoxin (3 mM), 10 μg CODH (3 mg/ml) and 30 μl NAD$^+$ (100 mM) under a CO atmosphere (solid square). $^{22}$Na$^+$ transport was not observed in the absence of NAD$^+$ (solid triangle). These data points represent two different biological replicates.

($E'_{O\,80\,°C} = -453$ mV[30]) and reduces NAD ($E_O' = -320$ mV). This exergonic reaction ($\Delta G^{o\prime} = -34.7$ kJ/mol) is supposed to be coupled to ion transport across the membrane[18]. The ion potential then drives the synthesis of ATP via an ATP synthase. Na$^+$ transport coupled to Fd$^{2-}$:NAD$^+$-oxidoreductase activity was indeed demonstrated in inverted membrane vesicles of *A. woodii*[10]. As expected, the reaction is reversible and an ion gradient is also used as driving force for the endergonic reduction of ferredoxin with NADH as reductant[31]. This activity is essential for nitrogen reduction of *R. capsulatus*, but also in *C. ljungdahlii*[20,32]. In addition, growing cells of *A. woodii* require reduced ferredoxin for biosynthetic reactions, but during growth on several substrates, only NAD is reduced. Under these conditions the Rnf complex is used to drive the endergonic reduction of ferredoxin with NADH as reductant at the expense of the electrochemical ion gradient[33].

Genetic experiments in which deletion of *rnf* genes led to a loss of Fd$^{2-}$:NAD$^+$-oxidoreductase activity together with the beforementioned experiments at inverted membrane vesicles are convincing circumstantial evidence that the Rnf complex is a respiratory enzyme[33]. However, the final proof requires the purification of the complex and its functional reconstitution into

liposomes. Despite several attempts in different laboratories, the Rnf complex resisted purification. Here, we describe the purification of the membrane-integral Rnf complex from the thermophile *T. maritima* and it is tempting to speculate that the thermophilic nature of the target protein was advantageous for its purification. Although Rnf activity was determined under suboptimal conditions at low temperatures, due to the instability at high temperatures of the auxiliary enzymes required for Fd reduction, we could unequivocally demonstrate $Na^+$ dependence of and, more important, $Na^+$ transport by the purified complex. This is, to our knowledge, the first proof that the Rnf complex is indeed an ion ($Na^+$) translocating respiratory enzyme.

Notably, the B subunit of the Rnf complex from *T. maritima* is much smaller than the B subunit from *A. woodii* or other

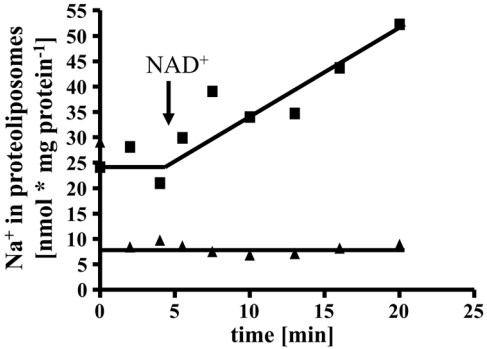

**Fig. 8 The effect of ionophores on $^{22}Na^+$ transport by the purified Rnf complex reconstituted into liposomes.** Proteoliposomes (protein concentration 0.99 mg/ml) in 1 ml 100 mM Tris-HCl, 100 mM malic acid, pH 7.4, containing 5 mM MgCl$_2$, 2 mM $^{22}$NaCl, 2 mM DTE and 4 µM resazurin were incubated with 40 µM TCS (solid square) or 40 µM ETH2120 (solid triangle) for 30 min at room temperature before starting $^{22}Na^+$ transport by addition of 10 µl ferredoxin (3 mM), 10 µg CODH (3 mg/ml) and 30 µl NAD$^+$ (100 mM) under a CO atmosphere. These data points represent two different biological replicates.

bacteria. Inspection of RnfB subunits from different sources revealed a striking difference in the length of subunit B (Fig. 9). Apparently, the difference is in the C terminus that harbors the FeS cluster. *Acetobacterium woodii* has six predicted FeS centers, *C. ljungdahlii*, has four, *R. capsulatus*, *Vibrio cholerae*, *Escherichia coli* have two, whereas *T. maritima* has only one. Despite these differences, ferredoxin from *C. pasteurianum* is able to deliver electrons to the Rnf complex. Whether the number of 4Fe-4S-cluster is indeed only one remains to be established.

The demonstration of an inhibition of the Rnf complex by DCCD is unprecedented. DCCD and $Na^+$ compete for a common binding site, as in $Na^+$-F$_1$Fo ATP synthases[27,28]. This may be a useful tool in the future to identify the $Na^+$-binding site. Radioactively labeled DCCD is no longer commercially available and a fluorescent analog, NCD4, did not inhibit the Rnf complex specifically. For the homologous Nqr complex Juárez et al.[34] speculated that the glutamate/aspartate in NqrB (E144, D397), in NqrE (E95) and in NqrD (D133) are potential $Na^+$-binding sites. The deletion of these four amino acids resulted in a reduced stimulation of ubiquinone reduction by $Na^+$ and mutations in NqrE-E95 resulted in an inhibition of electron transfer[34,35]. Except for NqrB-E144 all these amino acids are also conserved in the corresponding Rnf subunits from *T. maritima*, *A. woodii* and *V. cholerae* (Supplementary Figs. 7–9). However, there are more conserved acidic amino acids (NqrB-E274, NqrB-D346, NqrB-E380 NqrD-E119 and NqrD-E152) that could also be candidates for $Na^+$ binding. From the crystal structure of the Nqr complex, it was postulated that NqrB forms an ion channel and the D346, F342, F338, V161, I164 and L168 are involved in $Na^+$ translocation[23]. In the corresponding Rnf subunit of *T. maritima* RnfD only RnfD-D255 (*Vc.* NqrB-D346), RnfD-F252 (*Vc.* NqrB-F342) and RnfD-V79 (*Vc.* NqrB-V161) are conserved (Supplementary Fig. 9). This would lead to the assumption that RnfD would also form such an ion channel, but this has to await structural data.

The F$_1$F$_O$ ATP synthase of *T. maritima* not only has a conserved $Na^+$ binding motif in subunit *c* but its activity is $Na^+$ dependent and ATP hydrolysis is coupled to $Na^+$ transport. This is convincing evidence that the ATP synthase uses $Na^+$ as

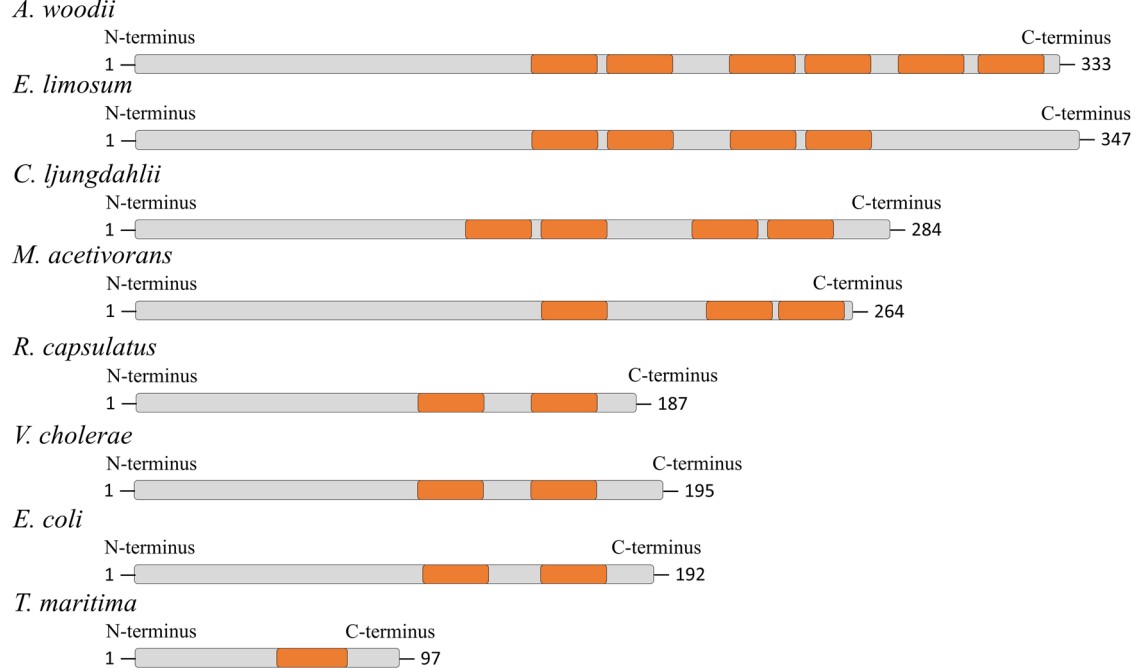

**Fig. 9 Comparison of RnfB subunits from different organisms.** The cysteine motifs (orange) were identified by "InterProScan" and each motif coordinates one 4Fe-4S cluster. *E. Eubacterium, M. Methanosarcina.*

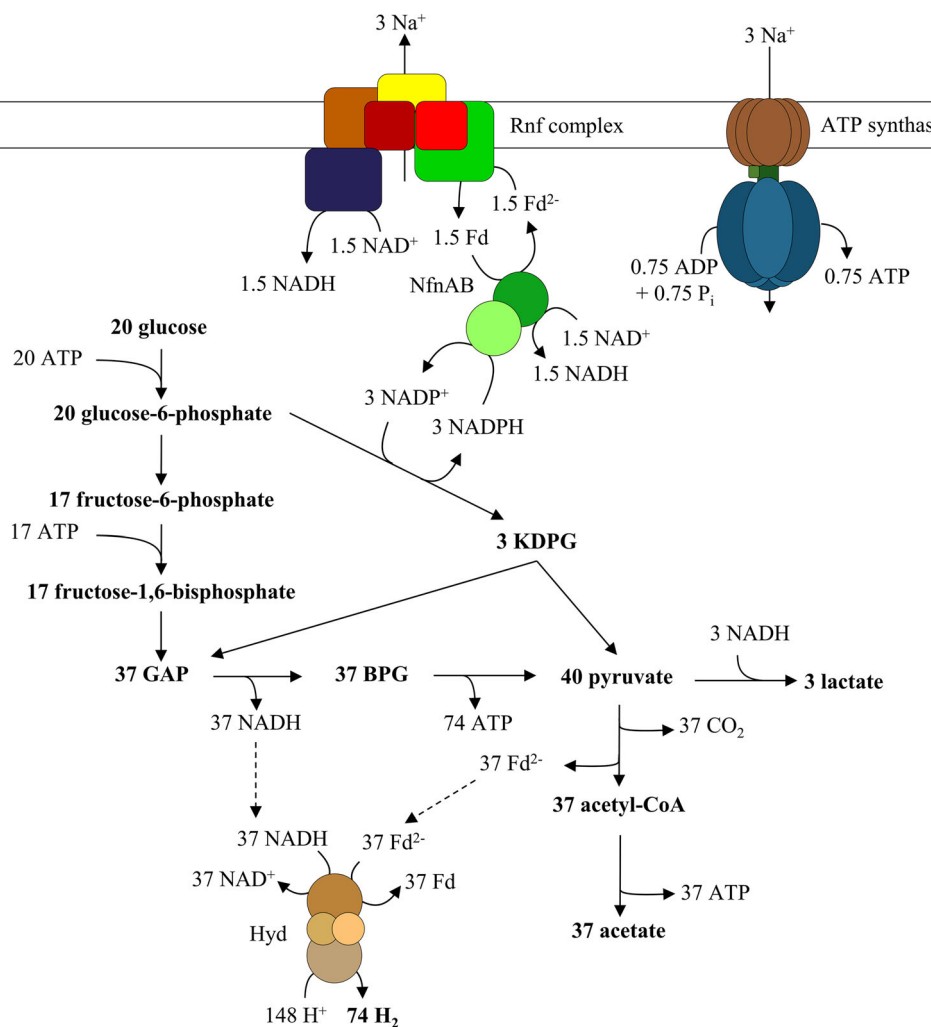

**Fig. 10 A model of glucose metabolism in *T. maritima*.** Glucose is oxidized via glycolysis and Entner-Doudoroff pathway and acetate is produced. NADPH generated in the glucose-6-phosphate dehydrogenase reaction is oxidized with an NADH-dependent $Fd^{2-}$:$NADP^+$-transhydrogenase (Nfn) by reducing Fd and NAD. Electrons from $Fd^{2-}$ are shuffled to NAD by the Rnf complex, thereby generating an $Na^+$ gradient that is used by $F_1F_O$ ATP synthase to generate ATP. NADH generated by the Rnf complex and Nfn is used to reduce pyruvate to lactate. All other reducing equivalents are used by the electron-bifurcating/confurcating hydrogenase (Hyd) to generate $H_2$. KDPG 2-keto-3-desoxy-6-phosphogluconate, GAP glyceraldehyde 3-phosphate, BPG bisphosphoglycerate. Note that the ion/electron and ion/ATP stoichiometries are based on thermodynamics (see also ref. [39]).

coupling ion, which is expected if the Rnf complex is $Na^+$ translocating. Apparently, the respiratory chain of *T. maritima* consists of Rnf and ATP synthase, which are connected by an electrochemical $Na^+$ potential.

*T. maritima* is a fermenting organism that converts sugars according to[36]

$$1\,\text{glucose} \rightarrow 2\,\text{acetate} + 2\,CO_2 + 4\,H_2. \quad (1)$$

Oxidation of glucose via the ED pathway generates NADPH that can be reoxidised via the electron-bifurcating transhydrogenase Nfn in connection with the Rnf complex (Fig. 10, ref. [37]). According to the model presented in Fig. 10, 3 mol of lactate are also produced to balance out the electrons; indeed, lactate production was seen in resting cell experiments[38]. Altogether, 3.7 ATP are generated by substrate level phosphorylation and additional 0.035 ATP are generated by electron transport phosphorylation with Rnf and ATP synthase. This is only about 1% of the total amount of ATP produced. This underlines the notion that, during chemoorganoheterotrophic growth, the function of the Rnf complex is not primarily in energy conservation but is that of a transhydrogenase essential to balance out the electrons[8]. It should be mentioned in this connection

that *T. maritima* has a membrane-bound, possibly ion translocating hydrogenase that could reduce protons (Supplementary Fig. 10) or $H_2S$ (Supplementary Fig. 11)[39]. The amount of ions translocated by the Rnf complex stays the same, but with protons as electron acceptors twice as much $Na^+$ would be translocated by the hydrogenase, giving 0.1 ATP/mol glucose. Based on theoretical considerations, reduction of S to $H_2S$ at high hydrogen partial pressures is increased since the hydrogenase is limited and more electrons flow towards sulfur reduction. This could increase the ATP yield by chemiosmosis to 1.79 per mol glucose, already 46% of the yield obtained by SLP.

## Methods

**Growth of *T. maritima*.** Cells of *T. maritima* (DSM3109) were grown at 80 °C in a 100 l fermenter under anoxic conditions as described previously with starch as substrate[38]. Harvested cells were frozen and stored at −80 °C.

**Purification of the Rnf complex and the ATP synthase.** In order to prevent protein damage through oxygen, all preparations and purification steps were performed under anoxic conditions in an anaerobic chamber (Coy Laboratory Products, Gras Lake Charter Township, MI). Additionally, all buffers contained 2 mM dithioerythritol (DTE), 4 μM resazurin and 5 μM FMN. Cells were resuspended and washed in lysis buffer (50 mM Tris-HCl, pH 8, 20 mM MgSO₄).

Sedimented cells were resuspended in lysis buffer containing 0.5 mM phenylmethylsulfonyl fluoride (PMSF) and 0.1 mg/ml DNaseI and disrupted by a single passage through a French Press with 1000 psig (100 bar). Cell debris were removed by a centrifugation step ($24,000 \times g$, 20 min, 4 °C). With the following ultracentrifugation step ($130,000 \times g$, 90 min, 4 °C), cytoplasmatic and membrane fractions were separated. Membranes were washed three times in lysis buffer and sedimented again through ultracentrifugation ($130,000 \times g$, 30 min, 4 °C). As the yield of the Rnf complex should be maximized, cell debris were not removed before purification. An ultracentrifugation step ($130,000 \times g$, 60 min, 4 °C) followed the cell disruption directly. Sedimented cells debris and membranes were washed three times with lysis buffer and sedimented again through ultracentrifugation ($130,000 \times g$, 30 min, 4 °C). Incubating the sample with DDM for 16 h at 4 °C led to a complete solubilization of the Rnf complex. In order to dilute the DDM concentration, solubilized proteins were mixed with lysis buffer in a ratio of 1:1. Solubilized proteins were applied to a Q-sepharose™ High Performance column ($1.6 \times 12$ cm) equilibrated with buffer A (50 mM Tris-HCl, pH 8, 20 mM MgSO$_4$, 0.02% DDM). A washing step using 100 mM NaCl using buffer B (50 mM Tris-HCl, pH 8, 20 mM MgSO$_4$, 0.02% DDM, 1 M NaCl) followed. Elution was performed by applying a linear gradient over 200 ml from 0 mM to 350 mM NaCl. A ferredoxin-dependent NAD$^+$ reductase eluted at approximately 230 mM NaCl. Pooled fractions were concentrated by ultrafiltration in 100-kDa Vivaspin tubes (Satorius Stedim Biotech GmbH) and applied to a Sephacryl™ S-300 HP ($1.6 \times 60$ cm) column equilibrated with buffer C (50 mM Tris-HCl, pH 8, 20 mM MgSO$_4$, 0.02% DDM, 150 mM NaCl). The Fd$^{2-}$:NAD$^+$-oxidoreductase (Fno) eluted in a single peak and was concentrated as performed previously and stored at 4 °C. For some experiments, it was necessary to remove NaCl by using a HiTrap-Desalting column where the buffer was changed to sodium free buffer C.

**Preparation and purification of the Rnf complex**. All steps until the solubilization were performed as described above. Resuspended membranes were used for protein solubilization with DDM. One milligram DDM was added per 1 mg protein and incubated for 2 h at room temperature under stirring. Solubilized membranes were removed by ultracentrifugation (45,000 rpm, 45 min, 4 °C). Solubilized proteins were precipitated with polyethylene glycol 6000 (PEG6000). For the first precipitation step, 7% PEG6000 [v/v] was added and incubated for 30 min at 4 °C under stirring. Precipitated proteins were removed by ultracentrifugation (38,000 rpm, 30 min, 4 °C). For the second precipitation step, the PEG6000 concentration was increased to 12% PEG6000 [v/v]. Precipitated proteins were collected by ultracentrifugation (38,000 rpm, 30 min, 4 °C) and resuspended in buffer A. Resuspended proteins were loaded onto a sucrose density gradient (20–50%) and centrifuged for 20 h in a vertical rotor (Beckman Optima L90-K, VTi50 rotor; 38,000 rpm, 4 °C). Sucrose gradients were fractionized (Ismatec IPC, Ismatec, Wertheim, Germany) and in each fraction Fno activity was measured as described later. Fractions with the highest Fno activity were pooled and applied to an anionic exchange chromatography column using Q-sepharose™ (GE Healthcare, Little Chalfont, UK) equilibrated with buffer A. Proteins were eluted with an NaCl gradient in buffer B at a flowrate of 2 ml/min. A ferredoxin-dependent NAD$^+$ reductase eluted at approximately 100 mM NaCl. Fractions with the highest Fno activity were pooled and applied to a biionic exchange chromatography column using Hydroxyapatite (Bio-Rad, Hercules, CA, USA) equilibrated with buffer D (5 mM KPO$_4$, 150 mM NaCl, 2 mM DTE, 4 μM resazurin, 5 μM FMN, 0.02% DDM [w/v], pH 7.0). Protein was eluted with a potassium phosphate gradient (153.5 to 500 mM KPO$_4$) with buffer E (buffer D with 500 mM KPO$_4$, pH 7.0) with a flowrate of 2 ml/min. Fractions with the highest Fno activity were pooled and concentrated (molecular mass cutoff 100 kDa) and applied to a gel filtration using Sephacryl S300 (GE Healthcare, Little Chalfont, UK) equilibrated with buffer F (buffer A containing 250 mM NaCl, pH 8.0) with a flowrate of 0.5 ml/min. The Fno activity eluted after 62 ml. The size of the protein was determined using a high molecular weight calibration kit (GE Healthcare, Little Chalfont, UK) under identical conditions. The enzyme was stable for 1 week when stored at 4 °C. For measurements in the absence of Na$^+$ the Sephacryl S300 pool was concentrated and applied on a desalting chromatography column using HiTrap-Desalting column (GE Healthcare, Little Chalfont, UK) equilibrated with buffer A with a flowrate of 1 ml/min. Fractions with Fno activity were pooled.

**Measurement of Rnf activity**. Fd$^{2-}$:NAD$^+$-oxidoreductase activity was measured as described[31] at 60 °C with 3 mM NAD$^+$. NaCl concentrations were adjusted and are indicated. Membranes or purified protein were used as sample. Determination of the temperature optimum was done by measuring the NADH:MV$_{ox}$ oxidoreductase activity of the Rnf complex since ferredoxin from *C. pasteurianum* and ACS/CODH from *A. woodii* are inactive at higher temperatures. These measurements were performed in anoxic cuvettes sealed with rubber stoppers and an atmosphere consisting of 100% N$_2$ ($0.5 \times 10^5$ Pa). They contained 1 ml buffer (20 mM Tris-HCl, pH 7.7, 2 mM DTE, 4 μM Resazurin, 20 mM NaCl) and 10 mM methylviologen. After adding the sample and an incubation time at the according temperature, the enzymatic reaction was started by adding 2 mM NADH. The formation of reduced methylviologen was measured photometrically at 604 nm ($\varepsilon_{MV} = 13.9$/mM/cm).

**Measurement of ATPase activity**. ATPase activity was measured in buffer containing 100 mM Tris-HCl, 100 mM maleic acid, 20 mM NaCl, 5 mM MgCl$_2$ at 80 °C. The pH was adjusted to 7.4 with KOH. The sample was preincubated at the given temperature before addition of 3 mM Na$_2$-ATP to start the reaction. ATP-dependent formation of inorganic phosphate was followed as described[40]. Determination of Na$^+$-dependence was carried out using NaCl-free buffer and K$_2$-ATP.

**Inhibition assays**. The samples were preincubated with DCCD at room temperature with and without 20 mM NaCl for 20 and 5 min at 60 °C before starting the reaction by adding the substrate. NaCl was added with the start of the reaction. DCCD was dissolved in ethanol; controls received the solvent only.

**In-gel activity assays**. Enzyme activity could also be shown in polyacrylamide gels. Rnf activity was demonstrated as an NADH:3-(4,5dimethylthiazol-2-yl)-2,5-diphenyltetrazolium bromide (MTT) oxidoreductase activity[41]. A native polyacrylamide gel was incubated in anoxic buffer (20 mM Tris-HCl, pH 7.7, 2 mM DTE, 20 mM NaCl) containing 200 μM MTT and 2 mM NADH. After short incubation MTT was reduced irreversibly and formazan accumulated in the gel[41]. ATPase activity was also shown in a native PAGE gel. The gel was incubated for an hour in buffer containing 35 mM Tris-HCl, pH 8.3, 270 mM glycine, 14 mM MgSO$_4$, 0.2% (w/v) Pb(NO$_3$)$_2$ and 8 mM Na$_2$-ATP at 65 °C. Released inorganic phosphate formed a white precipitate with lead nitrate. Incubating the gel in 50% methanol (v/v) stopped the reaction.

**Determination of Na$^+$ concentration**. The amount of Na$^+$ was determined with an Orion 84-ROSS sodium electrode (Thermo Electron Corp. Witchford, UK) as described[42].

**Preparation of proteoliposomes**. Liposomes were prepared from *E. coli* total lipid extract as described[43]. Reconstitution of proteins into liposomes was carried out using a modified protocol[43]. Proteoliposomes containing the Rnf complex had to be prepared under anoxic condition. Liposomes with a uniform diameter were diluted in transport buffer (100 mM Tris-HCl, pH 7.4, 100 mM maleic acid, 5 mM NaCl, 5 mM MgCl$_2$) which was purged with N$_2$ and contained 2 mM DTE and 4 μM resazurin for measurements with the Rnf complex. After destabilization of the liposomes, protein was added in a protein to lipid ratio of 1:10 (w/w). Further steps were carried out as described[43].

**Measurement of $^{22}$Na$^+$ translocation**. Measurement of $^{22}$Na$^+$ translocation by the Rnf complex had to be performed under anoxic conditions. The buffer (100 mM Tris-HCl, pH 7.4, 100 mM maleic acid, 5 mM NaCl, 5 mM MgCl$_2$) contained additional 2 mM DTE and 4 μM resazurin and was purged with N$_2$. All transport studies were performed at 45 °C. Rnf batches were prepared in 3.5 ml glass vials containing 500 μl sample with a protein concentration of 1.4 mg/ml. Ferredoxin purified from *C. pasteurianum* according to ref. [44] was added to a concentration of 30 μM. The gas phase was adjusted to $0.5 \times 10^5$ Pa CO and carrier-free $^{22}$Na$^+$ was added to a final concentration of 1 μCi/ml. After incubation for 30 min at room temperature, ACS/CODH purified from *A. woodii* according to ref. [31] was added (30 μg/ml). The batch was incubated another 5 min at 45 °C to ensure ferredoxin reduction and temperature equilibration. Addition of 3 mM NAD$^+$ started the reaction. Measurement of $^{22}$Na$^+$ translocation catalyzed by the ATP synthase was performed under oxic conditions. 3.5 ml glass vials were filled with 500 μl sample (protein concentration corresponded to 0.66–1.4 mg/ml). Carrier-free $^{22}$Na$^+$ was added to the batch with final concentration of 1 μCi/ml. After an incubation time of 35 min (30 min at room temperature and 5 min at 45 °C), the reaction was started with the addition of 3 mM K$_2$-ATP. Samples (40 μl) of both batches were removed and Na$^+$ concentration was determined as described[43].

**Analytical methods**. The protein concentration content of non-membranous samples was determined according to ref. [45], that of membranes according to ref. [46]. Determination of protein concentration in proteoliposomes was performed as described[47]. Proteins were separated in 12% SDS polyacrylamide gels according to ref. [48]. Native PAGE was performed as described[49]. Proteins were stained with either Coomassie Blue[50] or silver[51]. Western blotting was done as described[52]. LC-MS/MS and MALDI measurements were performed by the "Functional Genomics Center Zürich".

**Statistics and reproducibility**. One biological replicate is defined as one Rnf/ATPase complex or Rnf complex purification from one batch of *T. maritima* cells. With $n \geq 3$ biological triplicates, all data points were taken and were used to calculate their mean with standard deviation SEM.

**Reporting summary**. Further information on research design is available in the Nature Research Reporting Summary linked to this article.

## Data availability

The authors declare that all data supporting the findings of this study are available within the article, Supplementary Information, and Supplementary Data 1. Source data underlying plots shown in figures are provided in Supplementary Data 1. The data that support the findings of this study are available on request from the corresponding author V.M.

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

## Acknowledgements

This work was supported by a grant (MU 801/16-1) from the Deutsche Forschungsgemeinschaft. Open access funding provided by Projekt DEAL.

## Author contributions

M.K. planned and carried out the experiments and interpreted the data. D.T. planned and carried out the experiments, interpreted the data and prepared the manuscript. H.H. supplied the *T. maritima* cells for the experiments. V.M. planned the experiments, interpreted the data and prepared the manuscript.

## Competing interests

The authors declare no competing interests.
