## [Peer Review File · Communications Biology]

Reviewers' comments:

Reviewer #1 (Remarks to the Author):

This paper describes the first purification of any RNF complex and its reconstitution in liposomes for Na⁺ pumping activity measurements. The selection of a thermophilic organism made it possible to have a successful enzyme preparation. This paper represents an important step in understating the function of this widespread complex.

Furthermore, the data on the coupling mechanisms with the synthesis of ATP is well documented. This paper represents a significant contribution. The methods are precise, and the statistical analysis of the data is satisfactory.

Some points for clarification:

P.5 (112-116): According to results in Fig.1, RNF can make functional dimers. What is the significance of this finding?

P.6-7 (145-155): Fig. 2a shows that the RNF activity with 20 mM NaCl is about 1.0 U/mg. But the corresponding condition in Fig. 2b showed a very different value, 2.0 U/mg. Please clarify this. Also, the data on the protection with LiCl and KCl seem to be missing.

P.7 (159-161): The DCCD inhibition data are missing.

P.8 (190-193): The authors mentioned that the 230 kDa band in SFig. 4c is a dimer, but this is too small because the dimer is 320 kDa. Is this a typo?

P.8 (193-194): In SFig. 5, the figure legend says that the ion concentrations are 0-20 mM, but the figure has only the range of 0-9.5 mM. Also, the DCCD concentrations on the figure are not corresponding to the figure legend.

P.9 (197-199): The data of the ionophore experiments are missing.

A clarification: the authors said that ²²Na⁺ transport was inhibited by the Na⁺ ionophore ETH2120". While I think I understand what is meant by this, likely, the ionophore release Na⁺, so the built of the Na⁺ gradient is not observed. This would not be an inhibition of the transport activity or the transport enzyme. This needs to be clarified.

P.10 (231-234): This sentence should be revised as follows: A. woodii, has six predicted FeS centers, C. ljungdahlii, has four, P. (?)R. capsulatus, Vibrio cholerae [spelling], Escherichia coli have two, whereas T. maritima has only one.

P.11 (263-265): The lactate should be 0.15 mol per 1 mol glucose.

Fig.6, SFig. 10, SFig. 11: The Na⁺/ATP ratios in the reaction of ATP synthase are different in these models. Please clarify.

Reviewer #2 (Remarks to the Author):

The authors describe a procedure to purify the membrane-associated Rnf complex (reduced ferredoxin: NAD⁺ oxidoreductase) from a thermophilic bacterium, Thermotoga maritima, in complex with the membrane-associated ATP synthase. The purified Rnf-ATP synthase supercomplex was reconstituted into liposomes and its catalytic properties were analyzed. Furthermore, the Rnf complex

was separated from the ATP synthase and shown to catalyze Na⁺ transport. In the presence of reduced ferredoxin and NAD⁺, Na⁺ accumulated up to 2.5 fold. The stoichiometry, Na⁺ transported per electron transferred, was not determined.

This is an excellent biochemical study of this relatively novel energy conserving electron transport complex present in many strictly anaerobic bacteria and some archaea. I recommend publishing after the following comments have been considered: #

- The abstract begins with the following sentence: "rnf genes are widespread in bacteria but the function of the gene products is unknown". This sentence is misleading and must be omitted. There is a lot known about the function of the Rnf complex as outlined by the authors in the introduction and the discussion. It is correct, however, that ion transport coupled to Rnf activity has never been shown with a purified enzyme, as stated in the abstract. This is the actual novel in the manuscript. To be more informative the authors should also mention in the abstract, that accumulation of sodium ions in inverted vesicles was only somewhat higher than 2 fold and thus not sufficient to allow energy conservation.

- In the manuscript the words autotrophic and heterotrophic are used to indicate lithotrophic and organotrophic growth. Although nowadays many microbiologists use these terms synonymously, this is not correct. To make the point: E. coli was recently engineered such that the bacterium synthesized all of its carbon from CO₂, the bacterium grew autotrophically. The energy for autotrophic growth was provided lithotrophically by the reduction of nitrate with H₂, growth was thus chemolithoautotrophic. You have to be exact!

- It is stated that the dependence of Rnf activity on the NaCl concentration followed a Michaelis-Menten kinetic with a K_m of 0.55 ± 0.03 mM (line 144) (Fig. 2 A). This is misleading, without the addition of sodium ions the Rnf activity was 25% of the maximal activity in the presence of sodium ions. Therefore the given accuracy of the K_m (+/- 0.03 mM) makes no sense.

Reviewer #3 (Remarks to the Author):

All Authors: Marin Kuhn, Dragan Trifunović, Harald Huber, and Volker Müller

The Rnf complex is the prototype of a simple respiratory enzyme, evolved very early in the history of life, but kept during evolution and today found in many anaerobic, aerobic and facultative anaerobic bacteria. It is the evolutionary ancestor of the NADH: quinone-oxidoreductase (Nqr) found in some bacteria. The energy derived from this enzyme together with the Mrp-Mbx complex are proposed to provide sufficient ion-motive force to drive the Na⁺ dependent ATP synthase to synthesize ATP. In recent years, the Müller lab has contributed significantly to the evolutionary understanding of these classes of enzymes.

In the study presented by Kuhn et al, the authors demonstrated for the first time that the Rnf complex is a Na⁺ translocating enzyme which is essential for the Na⁺ dependent engine, ATP synthase. They developed new protocols to isolate both complexes in detergent and to reconstitute both enzymatically active enzymes. The data shed new light on the evolutionary variations of respiratory enzymes.

Before publication, the following points should be addressed:

1) Page 4, lines 91-92: "This accommodates only one predicted 4Fe-4S center,"
Is there evidence for such a statement? If so, please add the reference?

- 2) Page 5, lanes 99-100: "The Fd₂:NAD⁺-oxidoreductase was 97 mU/mg ...". The authors should add a figure in Supplementary data to this result.
- 3) Page 5, lanes 118-119: According to Nat. Comm. guidelines, the LC/MS/MS data should be included under Supplementary data. The same is the case for page 6, lanes 126-127.
- 4) Page 6, lanes 126-127: The LC/MS/MS and MALDI-TOF data should be included under Supplementary data.
- 5) Page 7, lane 152: like in case of "21 %", the author should write the value and unit as 21%, and should do this throughout the text.
- 6) Figure 1B, subunit delta is substoichiometric, which becomes obvious when compared with the band of subunit epsilon, which is unusual. The authors have to comment on this in particular because the ATPase activity gel in Figure 1a shows at least 2 bands representing the F1FO ATP synthase.
- 7) The grant number has to be added under Acknowledgments.

Dear Prof. Lee,

thank you for handling our manuscript (COMMSBIO-20-0660-T). We have paid regard to the comments of the reviewers as specified below and hope that our revised manuscript is now acceptable for publication. Changes in the manuscript suggested by reviewers are highlighted in yellow, typos, grammar corrected by us are highlighted in green.

For your convenience, the reviewer comments are given first, followed by our answer (→). The line numbering refers to the new version of the manuscript without marked changes, if not stated otherwise.

Reviewer #1 (Remarks to the Author):

This paper describes the first purification of any RNF complex and its reconstitution in liposomes for Na⁺ pumping activity measurements. The selection of a thermophilic organism made it possible to have a successful enzyme preparation. This paper represents an important step in understating the function of this widespread complex.

Furthermore, the data on the coupling mechanisms with the synthesis of ATP is well documented.

This paper represents a significant contribution. The methods are precise, and the statistical analysis of the data is satisfactory.

Some points for clarification:

P.5 (112-116): According to results in Fig.1, RNF can make functional dimers. What is the significance of this finding?

→ The reviewer is completely right, according to the apparent migration behavior in native gels Rnf dimers as well as monomers are possible. We have not elaborated on the functional relevance of a Rnf-dimer because we think that currently it is too early to be sure that the functional state of Rnf is dimeric; our data do not support this notion. It may well be that the possible dimer is a preparation artefact. Many more and different biophysical analyses are required to address this question. Therefore, at this point, we would like to restrain from a statement regarding the oligomeric state of the complex. See also below.

P.6-7 (145-155): Fig. 2a shows that the RNF activity with 20 mM NaCl is about 1.0 U/mg. But the corresponding condition in Fig. 2b showed a very different value, 2.0 U/mg. Please clarify this. Also, the data on the protection with LiCl and KCl seem to be missing.

→ Again, the reviewer is right, the values are different. This is due to the variation in activity of different badges of enzyme. Purification of the highly oxygen-sensitive enzyme takes at least 7 days and the yield is very low. From one purification, the amount of enzyme is hardly enough to make the experiments depicted in Fig. 2a. The experiment described in Fig. 2b were done with a different badge. We have added a sentence to the results section to explain this point. The data of the protection with LiCl and KCl have been added as new SFig. 5.

P.7 (159-161): The DCCD inhibition data are missing.

→ The data on the DCCD inhibition have been added as new SFig 6.

P.8 (190-193): The authors mentioned that the 230 kDa band in SFig. 4c is a dimer, but this is too small because the dimer is 320 kDa. Is this a typo?

→ No it is not a typo. But there is an apparent discrepancy between the apparent molecular masses of the complex in Fig. 1 and in SFig. 4C (now SFig. 7C). In Fig. 1 two complexes are clearly visible and, therefore, we feel that it is appropriate to refer to them as possible monomers and dimers, well knowing that the size doesn't fit exactly to the calculated masses. However, we now state in the text the mass of the potential dimer is within the experimental error for a dimer.

In new SFig. 7C only one complex is visible that smears over a range from 215-270 kDa with an average value of 242.5 kDa. This is actually in between a monomer (160 kDa) and a dimer (320 kDa). Therefore we leave the oligomeric state open, since it not important for our conclusion. We now state in the text: "Whether this represents a monomer or a dimer needs to be addressed in the future".

P.8 (193-194): In SFig. 5, the figure legend says that the ion concentrations are 0-20 mM, but the figure has only the range of 0-9.5 mM. Also, the DCCD concentrations on the figure are not corresponding to the figure legend.

→ Thanks for the comment. Both ranges were mixed up. That has been corrected.

P.9 (197-199): The data of the ionophore experiments are missing.

A clarification: the authors said that $^{22}\text{Na}^+$ transport was inhibited by the Na^+ ionophore ETH2120". While I think I understand what is meant by this, likely, the ionophore release Na^+ , so the built of the Na^+ gradient is not observed. This would not be an inhibition of the transport activity or the transport enzyme. This needs to be clarified.

→ The data of the ionophore experiments have been added as new SFig. 10. Again completely right. Sorry for this inprecise wording. It has been corrected.

P.10 (231-234): This sentence should be revised as follows: *A. woodii*, has six predicted FeS centers, *C. ljungdahlii*, has four, *P. (?)R. capsulatus*, *Vibrio cholerae* [spelling], *Escherichia coli* have two, whereas *T. maritima* has only one.

→ Thank you! It has been changed as suggested.

P.11 (263-265): The lactate should be 0.15 mol per 1 mol glucose.

→ Thank you! It has been corrected.

Fig.6, SFig. 10, SFig. 11: The Na^+/ATP ratios in the reaction of ATP synthase are different in these models. Please clarify.

→ Since the number of c subunits per ATP synthase monomer is not known, we assumed a value of 4 Na^+/ATP for our calculations. This gives the correct numbers in SFig. 10 and SFig. 11 (now SFig. 14 and SFig. 15). The number in Fig. 6 was indeed not correct, it was mixed up. It has been corrected to 0.75, based on the value given above.

Reviewer #2 (Remarks to the Author):

The authors describe a procedure to purify the membrane- associated Rnf complex (reduced ferredoxin: NAD^+ oxidoreductase) from a thermophilic bacterium, *Thermotoga maritima*, in complex with the membrane-associated ATP synthase. The purified Rnf-ATP synthase supercomplex was reconstituted into liposomes and its catalytic properties were analyzed.

Furthermore, the Rnf complex was separated from the ATP synthase and shown to catalyze Na⁺ transport. In the presence of reduced ferredoxin and NAD⁺, Na⁺ accumulated up to 2.5 fold. The stoichiometry, Na⁺ transported per electron transferred, was not determined.

This is an excellent biochemical study of this relatively novel energy conserving electron transport complex present in many strictly anaerobic bacteria and some archaea. I recommend publishing after the following comments have been considered:

- The abstract begins with the following sentence: "rnf genes are widespread in bacteria but the function of the gene products is unknown". This sentence is misleading and must be omitted. There is a lot known about the function of the Rnf complex as outlined by the authors in the introduction and the discussion. It is correct, however, that ion transport coupled to Rnf activity has never been shown with a purified enzyme, as stated in the abstract. This is the actual novel in the manuscript.

→ The first sentence has been changed accordingly.

To be more informative the authors should also mention in the abstract, that accumulation of sodium ions in inverted vesicles was only somewhat higher than 2 fold and thus not sufficient to allow energy conservation.

→ Yes, the reviewer is right that the accumulation is only 2-fold. However the driving force for ATP synthesis *in vivo* and *in vitro* is not the chemical Na⁺ potential but the electrochemical potential ($\Delta\Psi$) established by the transport of a positive charge. In general, due to the small electrical capacity of biological membranes, $\Delta\Psi$ is generated very fast by the movement of relatively small amounts of charges. This can also be seen in our manuscript by the stimulation of Na⁺ transport by addition of the protonophore TCS that dissipates the electrical field. Moreover, also *E. coli* does not have a significant ΔpH while grown at pH 7.0, and the $\Delta\Psi$ is the driving force for ATP synthesis. Therefore, the "small" accumulation factor of 2 does not argue against a role of Rnf in energy conservation. Other experiments such as genetic deletions have to be done to address this question and indeed, Rnf deletion strains of *A. woodii* are not able to grow under chemolithoautotrophic conditions.

- In the manuscript the words autotrophic and heterotrophic are used to indicate lithotrophic and organotrophic growth. Although nowadays many microbiologists use these terms synonymously, this is not correct. To make the point: *E. coli* was recently engineered such that the bacterium synthesized all of its carbon from CO₂, the bacterium grew autotrophically. The energy for autotrophic growth was provided lithotrophically by the reduction of nitrate with H₂, growth was thus chemolithoautotrophic. You have to be exact!

→ Changed, as suggested.

- It is stated that the dependence of Rnf activity on the NaCl concentration followed a Michaelis-Menten kinetic with a K_m of 0.55 ± 0.03 mM (line 144) (Fig. 2 A). This is misleading, without the addition of sodium ions the Rnf activity was 25% of the maximal activity in the presence of sodium ions. Therefore the given accuracy of the K_m (+/- 0.03 mM) makes no sense.

→ During the measurement without any addition of external NaCl the buffer still contained 0.12 mM Na⁺. Therefore the 25 % of the maximal activity was at 0.12 mM and not at 0 mM Na⁺; the x-scale is too small to see this, but this is written in the text. Anyway, we have changed from "0.55 ± 0.03 mM" to around 0.5 mM.

Reviewer #3 (Remarks to the Author):

All Authors: Marin Kuhn, Dragan Trifunović, Harald Huber, and Volker Müller

The Rnf complex is the prototype of a simple respiratory enzyme, evolved very early in the history of life, but kept during evolution and today found in many anaerobic, aerobic and facultative anaerobic bacteria. It is the evolutionary ancestor of the NADH: quinone-oxidoreductase (Nqr) found in some bacteria. The energy derived from this enzyme together with the Mrp-Mbx complex are proposed to provide sufficient ion-motive force to drive the Na⁺ dependent ATP synthase to synthesize ATP. In recent years, the Müller lab has contributed significant to the evolutionary understanding of these classes of enzymes.

In the study presented by Kuhn et al, the authors demonstrated for the first time that the RnF complex is a Na⁺ translocating enzyme which is essential for the Na⁺ dependent engine, ATP synthase. They developed new protocols to isolate both complexes in detergent and to reconstitute both enzymatically active enzymes. The data shed new light on the evolutionary variations of respiratory enzymes.

Before publication, the following points should be addressed:

1) Page 4, lanes 91-92: “This accommodates only one predicted 4Fe-4S center,”
Is there evidence for such a statement? If so, please add the reference?

→ There is no experimental evidence for this notion but we elaborate on this topic in the discussion and in new SFig. 10. As you will see there, the statement is based on sequence comparisons.

2) Page 5, lanes 99-100: “The Fd₂:NAD⁺-oxidoreductase was 97 mU/mg ...”.
The authors should add a figure in Supplementary data to this result.

→ The data has been added as new SFig. 1.

3) Page 5, lanes 118-119: According to Nat. Comm. guidelines, the LC/MS/MS data should be included under Supplementary data. The same is the case for page 6, lanes 126-127.

→ The LC/MS/MS and Maldi data have been added as STab. 2. The text has been changed accordingly.

4) Page 6, lanes 126-127: The LC/MS/MS and MALDI-TOF data should be included under Supplementary data.

→ The LC/MS/MS and Maldi data will be added as new STab. 2. The text has been changed accordingly.

5) Page 7, lane 152: like in case of “21 %”, the author should write the value and unit as 21%, and should do this throughout the text.

→ Thanks! This has been corrected.

6) Figure 1B, subunit delta is substoichiometric, which becomes obvious when compared with the band of subunit epsilon, which is unusual. The authors have to comment on this in particular because the ATPase activity gel in Figure 1a shows at least 2 bands representing the F_1F_0 ATP synthase.

→ The reviewer is completely right, δ is substoichiometric in our preparation and this may lead to the two F_1F_0 complexes shown in Fig. 1a. We have added a sentence to the Results section to note this point. However, as the reviewer knows, purification of such complexes enzymes preserving their subunit structure and the subunits in stoichiometric amounts is a hard task that was not in focus of this work. Here, the goal was to identify and characterize the ion-dependence of the enzyme.

7) The grant number has to be added under Acknowledgments.

→ The grant number has been added under Acknowledgments.

REVIEWERS' COMMENTS:

Reviewer #1 (Remarks to the Author):

The authors have changed the manuscript according to the suggestions/comments of the 3 referees. The current version of the manuscript is much improved and deserves to be published.